# Phytochemical Properties of Silk Floss Tree Stem Bark Extract and Its Potential as an Eco-Friendly Biocontrol Agent against Potato Phytopathogenic Microorganisms

**Abdulaziz A. Al-Askar**

Department of Botany and Microbiology, College of Science, King Saud University, P.O. Box 2455, Riyadh 11451, Saudi Arabia; aalaskara@ksu.edu.sa

**Abstract:** In the current study, the ethanolic extract of the stem bark of *Ceiba speciosa*, the silk floss tree (SFSB), was evaluated against various phytopathogenic microorganisms, including *Ralstonia solanacearum*, *Dickeya solani*, *Pectobacterium atrosepticum*, *P. carotovorum*, *Fusarium oxysporum*, and *Rhizoctonia solani*. At 300 μg/mL concentration, the SFSB extract exhibited the highest inhibition percentages of 83.33 and 86.67 for *R. solani* and *F. oxysporum*, respectively. In addition to its antimicrobial activity, SFSB extract exhibited strong antioxidant activity (IC50 value of 140.88 g/mL). HPLC analysis of the extract revealed the presence of various phenolic acids and flavonoids. Among these compounds, naringenin (18,698.83 μg/g), chlorogenic acid (2727.49 μg/g), ferulic acid (1276.18 μg/g), syringic acid (946.26 μg/g), gallic acid (812.34 μg/g), and methyl gallate (651.73 μg/g) were found to be the most abundant constituents. GCMS analysis showed that there were antimicrobial compounds like terpenoids, benzoic acid derivatives, phthalate esters, and different fatty acids. Isopropyl myristate was the most common compound, with a relative abundance of 55.61%. To our knowledge, this is the first investigation on the phytochemical composition and antimicrobial properties of SFSB extract. Consequently, utilizing SFSB extract could hold significant potential as a sustainable and natural approach for controlling and mitigating plant diseases.

**Keywords:** silk floss; extract; potato (*Solanum tuberosum*); antibacterial; antioxidant; antifungal; DPPH; GCMS; ITS; phylogenetic; HPLC

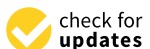



## 1. Introduction

Phytochemicals like flavonoids, alkaloids, tannins, and terpenoids are found in medicinal plants and are known for their antimicrobial and antioxidant properties [1]. Many plant species have been extensively studied for their antimicrobial effects. Therefore, natural substances have the potential to fight numerous plant diseases without causing any negative impact on the environment. Various plant extracts' antimicrobial and antifungal properties have become increasingly popular and scientific [2,3]. Crude extracts of herbs have been found to exhibit antimicrobial properties against a broad range of Gram-positive and -negative bacteria [4–6].

*Ceiba speciosa*, the silk floss tree, belongs to the Bombacaceae family, a group of flowering plants that includes approximately 200 species across 28 genera. These plants are perennial, deciduous, and woody trees that grow in tropical and subtropical regions, particularly in tropical America. However, the conservation status of *C. speciosa* may not be considered threatened or endangered on a global scale. Due to its relatively wide distribution across South America and adaptability to different habitats, the species might be categorized as least concern or not assessed by the International Union for Conservation of Nature (IUCN) Red List, although it may be protected in certain countries where it is native. In Egypt, the Bombacaeae family is represented by two genera, *Bombax* and *Ceiba*, which are mainly cultivated for their decorative and shading uses [7]. Silk floss trees have been traditionally used to treat various health disorders, such as headaches, fevers, diabetes,

diarrhea, and parasitic infections. Biologically, some *Ceiba* species have been reported to possess a wide range of valuable properties, such as hepatoprotective, anti-inflammatory, antioxidant, hypoglycemic, and cytotoxic effects, with high safety margins [8–13]. Behiry et al. [14] have noticed that the ethanolic leaf extract of *C. speciosa* contains flavonoids that exhibit free radical scavenging (DPPH) and antioxidant properties. Similarly, Nasr et al. [15] have found that different fractions of *C. speciosa* leaf and stem alcoholic extracts function as effective antioxidant agents, as demonstrated by their ability to scavenge DPPH.

Egypt ranked 12th in the world in potato production in 2021, with 6.9 million tons of potatoes produced [16]. Several pathogens affecting the production of potatoes, such as *Pectobacterium carotovorum* and *P. atrosepticum,* cause potato tuber blackening in the field, as well as soft rot of the tubers during storage and in the field, which results in significant losses. Identified as the primary pathogens affecting potatoes, the two *Pectobacteria* pose a significant challenge for producers in terms of control [17–19]. Also, the bacterial strain *Dickeya solani,* found to cause blackleg disease in potatoes, survives in the soil for years and can infect plants through wounds or natural openings, causing significant yield losses. Prevention and control measures include good agricultural practices, such as using certified seed potatoes, crop rotation, and avoiding wounds during planting, as well as judicious use of chemical control measures [20–22]. Immediate action should be taken if the infection is suspected, with consultation with agricultural extension or plant disease specialists [23,24].

Bacterial wilt caused by *Ralstonia solanacearum* is a major concern for the cultivation of several crops, including tomato, potato, eggplant, and pepper, in tropical, sub-tropical, and warm regions worldwide. *R. solanacearum* strains are categorized into five races and five biovars, with no clear correlation between them. Due to the wide range of hosts and the pathogen's ability to live in plant debris for different periods, controlling the disease is challenging. Despite attempts to develop resistant cultivars, there are currently no effective control strategies available for this disease [25–27].

There have been several approaches devised and implemented to manage the disease, but the utilization of protective chemicals poses a risk to the safety of ware tubers for human consumption. Some alternative options for controlling the diseases include the use of plant resistance elicitors or natural extracts, such as essential oils and hemp flower water, lemon grass, aloe vera, neem, and borax salt [17,28–32]. The antibacterial properties of plant extracts from various plant organs were investigated in several studies to determine their effectiveness against plant pathogenic bacteria and fungi [33–36]. The active antimicrobial components of these extracts were found to be small terpenoids and phenolic compounds [37]. Several studies have shown that certain botanicals from plants have inhibitory effects on plant pathogenic bacteria such as *R. solanacearum*, *P. carotovorum* [38], and *D. solani* [38–40].

*Rhizoctonia solani* is a plant pathogenic fungus that causes various diseases in crops, such as damping-off, root rot, and foliar blight. Several studies have investigated the antifungal properties of various extracts against *R. solani*, and the results have been promising [41,42]. *Fusarium*, an expansive genus belonging to the Ascomycota phylum, encompasses a multitude of species widely distributed in soils or intimately associated with plants. This genus consists of a complex assortment of species with several clonal lineages [43]. *Fusarium* has a global prevalence and is notorious for instigating severe vascular wilts and inducing decay in diverse plant components such as roots, stalks, cobs, seedlings, tubers, bulbs, and corms across a broad range of plant species [44]. Moreover, *Fusarium* species contribute to the development of postharvest dry rot and stem-end rot in potato crops during the active growing season [45]. The growth of *Fusarium oxysporum*, a specific species within the genus, is particularly favored by arid soil conditions [43]. Recently, there has been a growing interest in studying the potential of plant extracts, including *C. speciosa*, for controlling fungal diseases in plants, such as *R. solani* and *Fusarium* sp. [14,46]. The safety, environmental sustainability, and widespread public acceptance of plant extracts have made plant extracts a popular area of research for controlling plant diseases [47,48]. Therefore, this study aimed to isolate and molecularly identify potato pathogens, investi-

gate the antibacterial and antifungal properties of the ecofriendly ethanolic extract derived from the stem bark of silk floss (SFSB), as well as identify the phytochemical components present in the extract using HPLC and GC-MS analysis.

## 2. Materials and Methods

### 2.1. Source of Plant Bacterial Phytopathogens

The bacterial strains used in this study were *Ralstonia solanacearum*, *Dickeya solani*, *Pectobacterium atrosepticum*, and *P. carotovorum*, which were previously isolated from potato, molecularly identified by the 16SrRNA gene, and accessioned with GenBank numbers LN681200, MN598003, MG706146, and MN598002, respectively [49–51].

### 2.2. Plant Fungal Pathogens

Two fungi isolated from potato tubers were used. The isolation and purification of potato fungal pathogens involved several steps, including morphological identification and molecular identification. To begin, samples were collected from infected potato plants exhibiting dry rot, stem end rot, and black scurf symptoms associated with fungi. These samples were carefully collected, ensuring the preservation of fungal structures, and minimizing contamination. Isolation was carried out by culturing the collected samples on potato dextrose media (PDA) suitable for the growth of the isolated fungi. The media were supplemented with tetracycline antibiotics to promote the growth of the target pathogens while inhibiting the growth of other bacterial microorganisms [52]. After incubation at 25 °C, individual fungal colonies that exhibited typical morphological characteristics of *Rhizoctonia* [53] and *Fusarium* [54] species were selected for further analysis. Pure cultures were obtained by transferring single hyphal fragments from the colonies onto fresh media plates. This process was repeated until a pure culture of each pathogen was obtained.

Morphological identification involved careful examination of the fungal colonies, hyphal characteristics, spore formation, and other observable features using microscopy. Specific morphological traits, such as colony color, texture, and spore morphology, were noted and compared to established descriptions and keys for *Rhizoctonia* and *Fusarium* species [54,55]. In addition to morphological identification, molecular identification techniques were employed to confirm the identity of the isolated pathogens. This typically involved DNA extraction from the pure fungal cultures [56], followed by PCR amplification of the specific target internal transcribed spacer (ITS) region by ITS1 and ITS4 primers [57]. The amplification process of the DNA was performed according to Nishizawa et al. [58], then the amplified product was subjected to DNA sequencing by the Sanger-sequencing method at Macrogen Company (Seoul, Republic of Korea) to determine the genetic identity of the pathogens. The phylogenetic tree of the isolated fungal pathogens was constructed by the MEGA 11 software program [59].

### 2.3. Preparation of Silk Floss Extract

The stem bark of silk floss (SFSB) was gathered in Alexandria governorate, Egypt, and air-dried at 25 °C for a fortnight. The next step involved using a Moulinex AR1044 grinding mill (Moulinex S.A., Paris La Defense Cedex, France) to finely grind the sample into a powder. A total of 400 mL of 96% ethanol was used to soak one hundred grams of the powder for one week. The resulting ethanol mixture was filtered, and the extract obtained was concentrated using a rotary vacuum rotary evaporator RE-2010 (BIOBASE, Jinan, Shandong, China). All the tested concentrations used in the study were diluted with 10% dimethyl sulfoxide (DMSO) [48].

### 2.4. Antibacterial Activity of SFSB Extract

The Kirby–Bauer disc diffusion test involves placing bacteria on a solid growth medium plate and adding disks impregnated with the extract at different amounts (conc. 100, 200, 300, 400, 600, 800, 1000, 2000, and 3000 μg/mL) onto the plate. A positive control (amoxicillin 25 μg/disc) and a negative control (10% DMSO) were used in the assay. Fol-

lowing overnight bacterial growth, transparent regions surrounding the discs indicate that the extract or the antibiotic can hinder bacterial growth [60]. The assay was repeated thrice.

### 2.5. Silk Floss Stem Bark Extract (SFSB) Antifungal Activity

An extract of SFSB was examined for its potential to combat the tested fungi using the poisoned food technique [61]. Various concentrations of SFSB extract (25, 50, 100, 200, and 300 μg/mL) were blended with potato dextrose agar (PDA) dishes and compared to a fungicide (Norma Cu-98; the final concentration in media is 1 μg/mL) and negative control (PDA-supplement with 10% DMSO). Fungal circular discs (0.5 cm) were put in PDA plates and incubated at 25 °C for 5 days. The assay was repeated thrice. Growth inhibition (%) was used to determine the effect of SFSB extract on the hyphal growth of the fungus using the formula, Growth inhibition (%) = [(Ti − Tf)/Ti] × 100, where Ti refers to the length of the fungal growth in the control and Tf refers to the hyphal growth length in the treated plate [62].

### 2.6. SFSB Extract Antioxidant Activity

The scavenging free radical efficiency was evaluated using the technique from Shimada et al. [32]. A total of 100 mL of methanol was used to dissolve 3.94 mg of DPPH. A 100 μL methanol mixture was gradually added to 6 mL of each extract concentration from 62.5 to 1000 μg/mL. The mixture was stirred at room temperature for 30 min. Each absorbance value (AV) was recorded at 517 nm. The DPPH reduction efficiency was calculated according to the formula DPPH% = [(Dp − Dh)/Dp] × 100, where Dh represents the sample AV and Dp represents the control AV, which includes all necessary reagents except for the test sample. The activity was determined by comparing the $IC_{50}$ (50% DPPH scavenging) value of the sample with that of butylated hydroxytoluene (BHT) [63].

### 2.7. HPLC Analysis

All the relevant data about the HPLC instrument, column, mobile phase, flow rate, gradient program, detector, injection volume, column temperature, and standard compounds are set out in Table 1.

**Table 1.** HPLC parameters used in this study.

| Parameter | Value | Reference |
|---|---|---|
| HPLC Instrument | Agilent 1260 series (Agilent Technologies, GmbH, Boblingen, Germany) | |
| Column | (Eclipse C18) dimensions: 4.6 mm (diameter) × 250 mm (length) Particle size: 5 μm | |
| Mobile phase | $H_2O$ (A) and 0.05% $CF_3COOH$ in $CH_3CN$ (B) | |
| Flow rate | 0.9 mL/min | [64] |
| Gradient program | In this stepwise program, the mobile phase composition changes abruptly at specific intervals to facilitate the separation of the sample components. From 0 to 5 min, the mobile phase contains 80% solvent A and 20% solvent B. From 5 to 8 min, the mobile phase changes to 60% solvent A and 40% solvent B. From 8 to 12 min, the mobile phase remains at 60% solvent A and 40% solvent B. From 12 to 16 min, the mobile phase returns to the initial condition of 82% solvent A and 18% solvent B. From 16 to 20 min, the mobile phase remains at 82% solvent A and 18% solvent B. | |
| Detector | Multi-wavelength type monitored at 280 nm | |
| Injection volume | 5 μL | |
| Column temperature | 40 °C | |
| Analyzed compounds (standards) | 17 common phenolic and flavonoid components: apigenin, caffeic acid, catechin, chlorogenic acid, cinnamic acid, coumaric acid, daidzein, ellagic acid, ferulic acid, gallic acid, methyl gallate, naringenin, pyro catechol, quercetin, rutin, syringic acid, vanillin | |

*2.8. Gas Chromatography Mass Spectroscopy (GCMS) Analysis*

The equipment, column, carrier gas, flow rate, ionization energy, scan time, fragment range, injection quantity, injector temperature, column oven temperature, and identification of phytochemicals of the gas chromatography–mass spectroscopy (GCMS) instrument are set out in Table 2.

**Table 2.** GCMS conditions, parameters, and values were used in this study.

| Parameter | Value |
|---|---|
| Equipment | Agilent 7000D |
| Column | 5% Diphenyl/95% Dimethylpolysiloxan column, packed with HP-5MS capillary column (30 m in length $\times$ 250 μm in diameter $\times$ 0.25 μm in thickness) |
| Carrier Gas | Pure helium gas (99.99%) |
| Flow Rate | 1 mL/min |
| Ionization Energy | 70 eV |
| Scan Time | 0.2 s |
| Fragment Range | 40 to 600 $m/z$ |
| Injection Quantity | 1 μL (split ratio 10:1) |
| Injector Temperature | 250 °C (constant) |
| Column Oven Temperature | 50 °C for 3 min, raised at 10 °C per min up to 280 °C, and final temperature was increased to 300 °C for 10 min |
| Identification of phytochemicals | Based on the comparison of their retention time (min), peak area, peak height, and mass spectral patterns with those spectral databases of authentic compounds stored in the National Institute of Standards and Technology (NIST) library |

*2.9. Statistical Analyses*

The statistics were conducted using the CoStat program version 6.45 (CoHort software). The ANOVA test was utilized to analyze the data, and the Tukey post hoc method was employed to compare the means at the level of probability $p \leq 0.05$.

## 3. Results and Discussion

*3.1. SFSB Extract Bacterial Inhibitory Effect In Vitro*

Table 3 shows the results of an experiment evaluating the antibacterial activity of SFSB extract against four bacterial strains, including *R. solanacearum, D. solani, P. atrosepticum*, and *P. carotovorum* (Figure S1). The inhibitory activity of the extract was measured in terms of the diameter of the inhibition zone (mm) around the disc containing the extract. The data are presented as mean values of triplicates. The data show that the SFSB extract had varying degrees of inhibitory activity against the different bacterial strains tested. The inhibitory activity increased with increasing concentration of the extract in most cases. For *R. solanacearum, D. solani, P. atrosepticum*, and *P. carotovorum*, the maximum inhibition zones (IZs) were observed at concentrations of 3000, 400, 3000, and 2000 μg/mL, respectively, and with IZs ranging from 8.67 to 14.33 mm at the highest concentration of 3000 μg/mL. In contrast, the suppressive effects of SFSB against *P. carotovorum* were relatively lower, with IZs ranging from 8.33 to 12.67 mm at the highest concentration of 3000 μg/mL. The positive control (amoxicillin) demonstrated more potent inhibitory activity against *D. solani, P. atrosepticum*, and *P. carotovorum* in comparison to the negative control (10% DMSO), which had no effect.

To summarize, based on the findings, it can be inferred that the SFSB extract exhibits promising antibacterial properties against the tested bacterial strains, with different levels of effectiveness depending on the bacterial isolate and the concentration of the extract used. In this regard, a study conducted by Kausar et al. [65] found that the oil obtained from *C. speciosa* leaves (COL) displayed different levels of bacterial inhibition, with the highest sensitivity observed in *S. aureus* (25 mm), followed by intermediate activity against *Escherichia coli* (15 mm). At the same time, *S. typhi* showed no sensitivity at a concentration of 3.64 mg. Another study reported that two organic fractions obtained from the *C. insignis* leaf ethanolic extract had strong antibacterial activity against species of *Bacillus* [66]. The

*n*-hexane oily extract (*n*-HOE) obtained from the bark of *Bougainvillea spectabilis* showed the most robust efficacy against *P. carotovorum* and *D. solani*, with 12- and 12.33-mm inhibition zones, respectively, at a concentration of 4000 µg/mL. On the other hand, *Citharexylum spinosum*-*n*HOE exhibited lower activity [50]. Amoxicillin is an antibiotic from the penicillin group, and it primarily acts against Gram-positive bacteria. It had different bactericidal actions against our studied Gram-negative bacterial isolates. In the literature, it also exhibits some activity against certain Gram-negative bacteria [67]. Its effectiveness against Gram-negative bacteria is limited due to the structure of their outer cell membrane, which makes it harder for the drug to penetrate and exert its action. It is important to note that many Gram-negative bacteria have developed resistance mechanisms against penicillin-type antibiotics, including amoxicillin [68,69]. For this reason, amoxicillin is often used in combination with other drugs or prescribed as part of a broader-spectrum antibiotic regimen for infections caused by suspected or known resistant organisms [69]. However, further studies are necessary to identify the bioactive compounds responsible for the observed activity and to evaluate the extract's potential use as a natural antibacterial agent.

**Table 3.** Growth inhibition (mm) of silk floss tree stem bark extract against different plant pathogenic bacteria.

| Treatments | Inhibition Zone (mm) | | | |
|---|---|---|---|---|
| Conc. (µg/mL) | *Ralstonia solanacearum* | *Dickeya solani* | *Pectobacterium atrosepticum* | *Pectobacterium carotovorum* |
| 100 | 7.33 ± 0.24 [d] | 7.67 ± 0.47 [c] | 7.33 ± 0.24 [d] | 7.33 ± 0.24 [b] |
| 200 | 7.67 ± 0.94 [cd] | 7.67 ± 0.94 [c] | 7.33 ± 0.94 [d] | 7.67 ± 0.62 [b] |
| 300 | 8.00 ± 0.82 [cd] | 8.67 ± 0.47 [bc] | 7.33 ± 0.24 [d] | 8.33 ± 1.25 [b] |
| 400 | 8.67 ± 0.47 [cd] | 8.67 ± 1.18 [bc] | 7.33 ± 0.94 [d] | 8.33 ± 0.94 [b] |
| 600 | 8.67 ± 1.70 [cd] | 8.67 ± 0.47 [bc] | 7.33 ± 0.47 [d] | 8.33 ± 0.47 [b] |
| 800 | 9.33 ± 0.24 [bcd] | 9.00 ± 0.82 [bc] | 7.67 ± 0.47 [cd] | 8.67 ± 0.94 [b] |
| 1000 | 10.33 ± 0.24 [bc] | 9.33 ± 0.47 [b] | 9.33 ± 0.47 [bc] | 9.00 ± 0.00 [b] |
| 2000 | 12.00 ± 0.41 [ab] | 9.33 ± 0.47 [b] | 9.33 ± 0.94 [bc] | 9.00 ± 0.00 [b] |
| 3000 | 14.33 ± 0.94 [a] | 9.67 ± 1.25 [b] | 9.67 ± 1.25 [b] | 9.67 ± 1.89 [ab] |
| * Pc | 11.67 ± 0.85 [ab] | 13.00 ± 0.00 [a] | 14.33 ± 0.94 [a] | 12.67 ± 0.47 [a] |
| ** Nc | 00.00 ± 0.00 [e] | 00.00 ± 0.00 [d] | 00.00 ± 0.00 [e] | 00.00 ± 0.00 [c] |

Silk floss bark extract (row label spanning treatment rows)

In the presented data, the use of different letters superscripted to the data values within the same column indicates that these values are statistically significant at a probability level of 0.05%. * Pc = positive control (Amoxicillin 25 µg/disc), ** Nc = negative control (DMSO 10%).

## 3.2. Fungal Pathogens

### 3.2.1. Isolation and Identification

Samples collected from infected potato tubers were successfully isolated and purified on PDA media for two fungal isolates. The pure cultures of each pathogen were obtained by transferring individual hyphal fragments from selected colonies. Morphological identification involved careful examination of the fungal colonies, hyphal characteristics, and spore formation. *Rhizoctonia solani* exhibited distinctive features such as fluffy white mycelium and brown sunken lesions on infected plant tissues [70]. *Fusarium oxysporum* displays white to pinkish mycelium with microconidia and macroconidia production [71]. By combining morphological identification with molecular techniques, the isolated fungal pathogens were accurately identified, allowing for a comprehensive understanding of their characteristics and enabling further research on their pathogenicity and management strategies in potato crops.

### 3.2.2. Molecular Identification and Phylogenetic Analysis

For molecular identification, DNA was extracted from pure fungal cultures. A specific internal transcribed spacer (ITS) region was amplified using PCR and then subjected to DNA sequencing. The obtained DNA sequences were compared to reference sequences available in public databases using sequence alignment tools. The analysis confirmed the

genetic identity of *R. solani* AG-3 PT [55] and *F. oxysporum* [44,52] based on the similarities between their DNA sequences and the reference sequences. The sequences were deposited in GenBank under accession numbers OR116526 and OR116506, respectively.

To construct a phylogenetic tree, the DNA sequences of the isolated *R. solani* isolate RHS-295 and *F. oxysporum* isolate FO-94, along with related species, were aligned and analyzed using phylogenetic analysis MEGA 11 software. The phylogenetic trees depicted the evolutionary relationships between different fungal species, including *R. solani* and *F. oxysporum*, based on their genetic similarities and divergence (Figures 1 and 2). The phylogenetic analysis revealed the placement of *R. solani* and *F. oxysporum* within their respective clades and provided insights into their evolutionary relatedness to other fungal species (Figures 1 and 2). It also helped in understanding the phylogenetic diversity and relationships among different isolates of *R. solani* and *F. oxysporum*. The fungal isolation, morphological identification, molecular identification, and phylogenetic analysis provided a comprehensive understanding of the presence and genetic relationships of *R. solani* and *F. oxysporum* fungal pathogens in the studied potato plants. These results contribute to our knowledge of their pathogenicity and can assist in the development of effective management strategies for controlling these pathogens in potato crops.

### 3.3. Effect of SFSB Extract on the Fungal Pathogens

Table 4 shows the effect of different amounts of SFSB extract on the growth of *R. solani* and *F. oxysporum*. The fungal growth decreased as the extract concentration increased from 25 to 300 μg/mL. The findings indicated that SFSB was more effective in inhibiting the growth of *F. oxysporum* than *R. solani* at all tested concentrations (Figure S2). At 300 μg/mL, the SFSB extract inhibited *F. oxysporum* radial growth by 86.67%, and no significant differences were observed among the concentrations of 50, 100, and 200 μg/mL and the control. However, the effect on *R. solani* growth was lower at 25 μg/mL compared to the Norma Cu-98 fungicide (1 μg/mL). Interestingly, the positive control (Norma Cu-98 fungicide, 1 μg/mL) also showed inhibitory effects against both fungi, although at lower levels compared to the test compound. The negative control (DMSO 10%) did not show any inhibitory effects against the fungi. Several methods, such as chemical fungicides and biological control agents, have been reported to be effective in reducing crop damage caused by fungal infections [72,73].

**Table 4.** Growth inhibition percentage (%) of plant pathogenic fungi in response to silk floss bark extract.

| Treatments Conc. (μg/mL) | Inhibition Percentage (%) | |
| --- | --- | --- |
| | *Rhizoctonia solani* | *Fusariumoxysporum* |
| 25 | 28.10 ± 4.71 [d] | 82.38 ± 1.78 [c] |
| 50 | 39.05 ± 2.94 [c] | 83.33 ± 0.67 [bc] |
| 100 | 41.90 ± 0.79 [c] | 86.19 ± 0.41 [ab] |
| 200 | 70.48 ± 1.35 [b] | 86.19 ± 0.67 [ab] |
| 300 | 83.33 ± 2.43 [a] | 86.67 ± 0.59 [a] |
| * Pc | 63.33 ± 0.76 [b] | 83.81 ± 0.44 [abc] |
| ** Nc | 00.00 ± 0.00 [e] | 00.00 ± 0.00 [d] |

In the presented data, the use of different letters superscripted to the data values within the same column indicates that these values are statistically significant at a probability level of 0.05%. * Pc = positive control (Norma Cu-98 fungicide, 1 μg/mL), ** Nc = negative control (DMSO 10%).

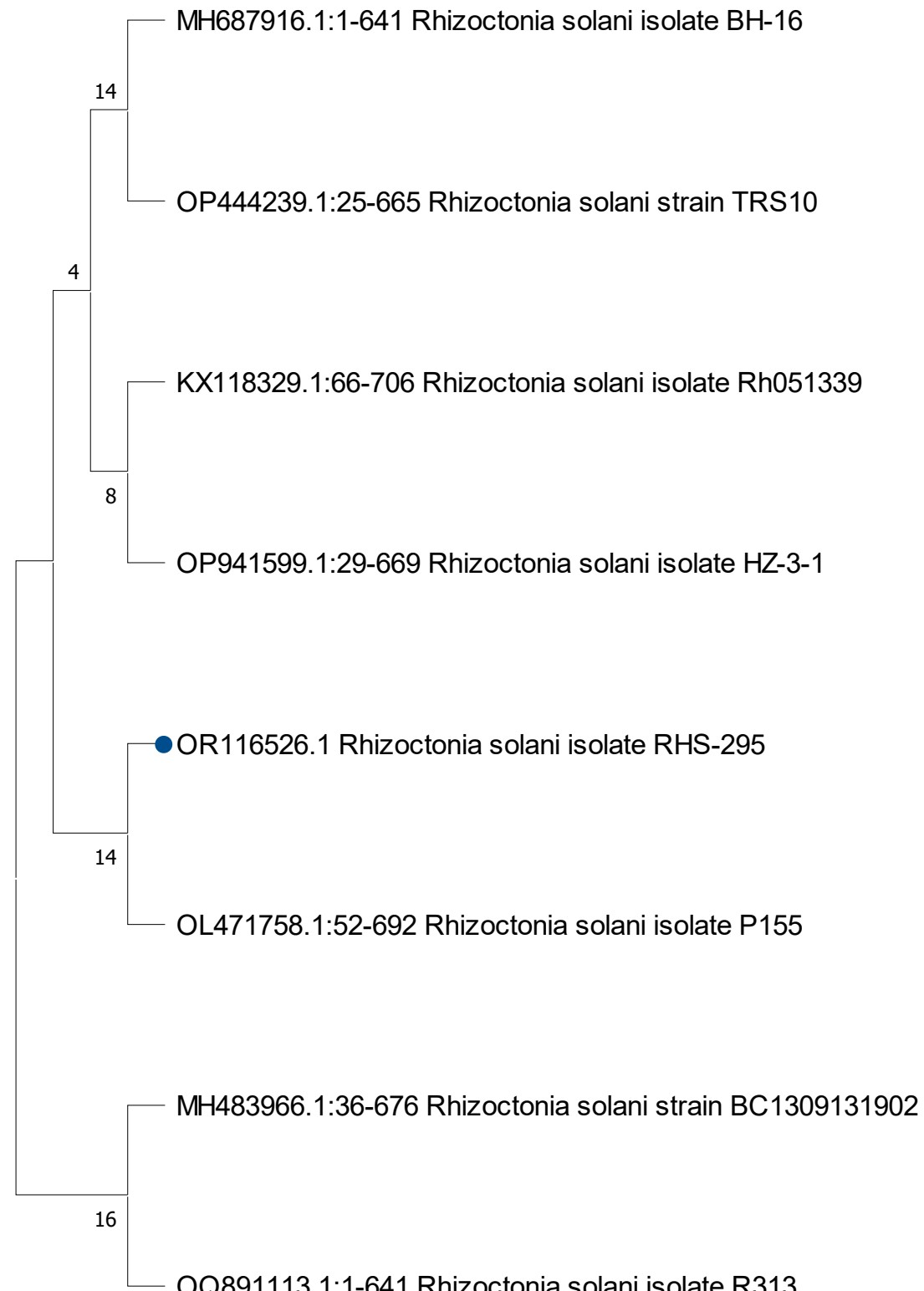

**Figure 1.** Maximum likelihood phylogenetic tree constructed based on ITS sequences of the studied isolates *Rhizoctonia solani* isolate RHS-295 (blue color) compared and aligned with other related genera delivered from NCBI GenBank portal.

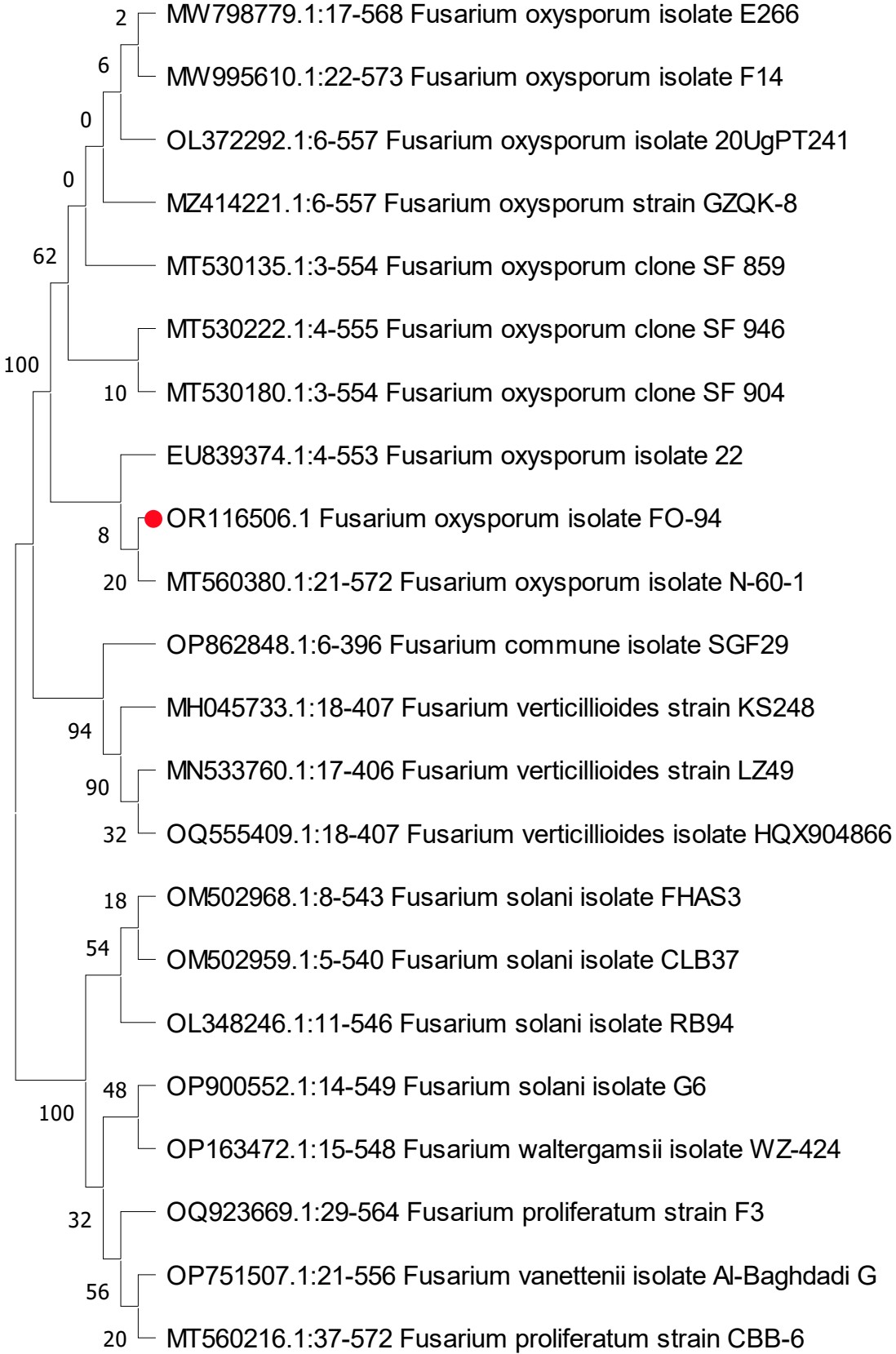

**Figure 2.** Maximum likelihood phylogenetic tree constructed based on ITS sequence of the studied *Fusarium oxysporum* isolate FO-94 (red color) compared and aligned with other related genera sequences delivered from NCBI GenBank portal.

Our current investigation showed a significant reduction in fungal development using SFSB extract. In a recent study conducted by the authors on COL extract antifungal activity against *R. solani* growth, the extract exhibited solid antifungal activity at low concentrations of 1–10 μg/mL [14]. In a previous study, the investigators found that 30 μg/mL of *Coccoloba uvifera* ethanolic extract inhibited *F. culmorum, R. solani,* and *B. cinerea* growth by 38.5%, 64.4%, and 100%, respectively [38]. At the same concentration of *R. solani* and *F. culmorum,* mycelia growth was also significantly inhibited by *Eucalyptus camaldulensis* hexane extract [46]. Similarly, *Acacia saligna* water extract suppressed *Fusarium* sp. and *R. solani* growth [74]. Previous research studies have shown that alcoholic extracts are more effective than water extracts in their antimicrobial properties [75–79]. Overall, the data suggest that the tested extract has the potential to be a natural fungicide against *R. solani* and *F. oxysporum.* Nevertheless, additional research is needed to fully explore the mechanism of action and potential application of the test compound in agriculture. Additionally, a comparison with commercial fungicides is needed to evaluate the effectiveness and practicality of the test compound in real-world scenarios.

*3.4. Antioxidant Activity*

The scavenging capacity of free radicals by SFSB extract was evaluated using the DPPH method, and the antioxidant activity value ($IC_{50}$) was found to be 140.88 μg/mL, compared to the $IC_{50}$ value of BHT of 5.96 μg/mL as illustrated in Figure 3. Our findings support earlier research that indicated *Ceiba* species have antioxidant properties. Also, our extract was rich in flavonoids, which act as antioxidants. Many flavonoids possess potent antioxidant activity due to their ability to scavenge free radicals, which can corrupt and wreck tissues [80]. The activity of flavonoids is due to their spatial arrangement of atoms and molecules, which allows them to donate electrons or hydrogen atoms to free radicals, thereby neutralizing their harmful effects. This mechanism of action is known as radical scavenging or free radical quenching [81].

Previous studies have reported high antioxidant activity for various fractions of leaf and stem ethanolic extracts from *C. speciosa* [15]. Refaat et al. [12] have also shown that the water, chloroform, and ethyl acetate extracts of different plant parts of other *Ceiba* spp. have remarkable scavenging antioxidant activities and high amounts of polyphenolics.

*3.5. Silk Floss (SFSB) Extract Polyphenolic Content*

The extract of SFSB was analyzed using HPLC, and the resulting data are presented in Figure 4 and Table 5. The analysis revealed several chemical compounds, with the predominant compounds (μg/g) being naringenin (18,698.83), chlorogenic acid (2727.49), ferulic acid (1276.18), syringic acid (946.26), gallic acid (812.34), methyl gallate (651.73), daidzein (628.91), vanillin (473.22), catechin (324.42), quercetin (220.42), ellagic acid (139.18), coumaric acid (104.47), rutin (97.16), and cinnamic acid (22.46). Some of the compounds, such as caffeic acid, pyrocatechol, apigenin, kaempferol, and hesperetin, were not detected in the extract (ND). It is worth noting that the presence or absence of a compound in an extract can depend on several factors, such as the extraction method used, the part of the plant used, and the maturity of the plant. Therefore, it is essential to consider these factors while interpreting the data. Flavonoids and polyphenols are commonly found in medicinal plants that have been used for many years. Antibacterial, antiviral, and antioxidant properties are exhibited by several phenolic acids, such as ellagic, chlorogenic, gallic, caffeic, and ferulic acids [82,83]. Flavonoids are a diverse group of phytochemicals that are found in various fruits, vegetables, and other plant-based foods. Many flavones, such as naringenin (NAR), have been shown to possess a variety of health-promoting properties, including antioxidant and anti-inflammatory properties. Several investigations have demonstrated the activity of NAR and its derivatives against bacteria and fungi. A study conducted by Duda-Madej et al. [84] has shown that it suppresses the growth of several bacterial strains, such as *S. aureus* and *Pseudomonas aeruginosa*.

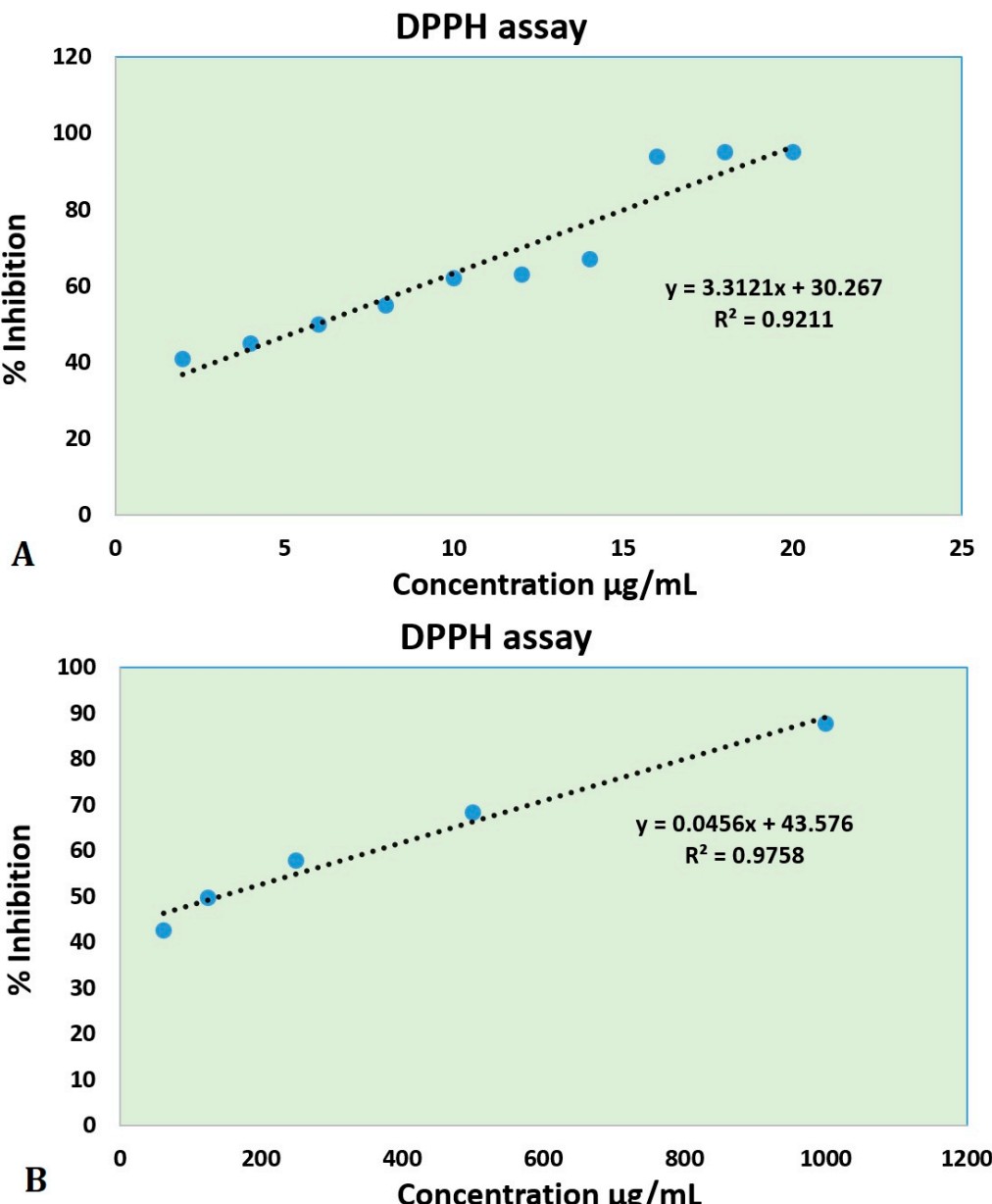

**Figure 3.** The scavenging activity of DPPH (IC$_{50}$) of butylated hydroxytoluene (BHT) (**A**) and the silk floss stem bark extract (**B**).

As a result, researchers are turning to natural compounds for their potential antibacterial activity, though natural compounds often show lower activity against G-negative bacteria. Despite being a well-known compound, naringenin has recently gained renewed interest due to its potential as an antibacterial agent. Researchers have measured the minimum inhibitory concentration (MIC) of NAR against various G-negative bacteria, including *E. coli*, *H. pylori*, *P. aeruginosa*, and *K. pneumoniae*, with values ranging from 0.5–1000 µg/mL. Studies have also demonstrated the NAR effect on genes involved in quorum sensing in *P. aeruginosa*, and its suppressive effect on *H. pylori* enzymes. While the results for NAR were significantly worse compared to some close derivatives of NAR, they showed promising MIC values [85–87]. Similar to other flavonoids, NAR inhibits fatty acid synthesis, disrupts gyrase activity, suppresses QS molecules, and enhances membrane permeability [88].

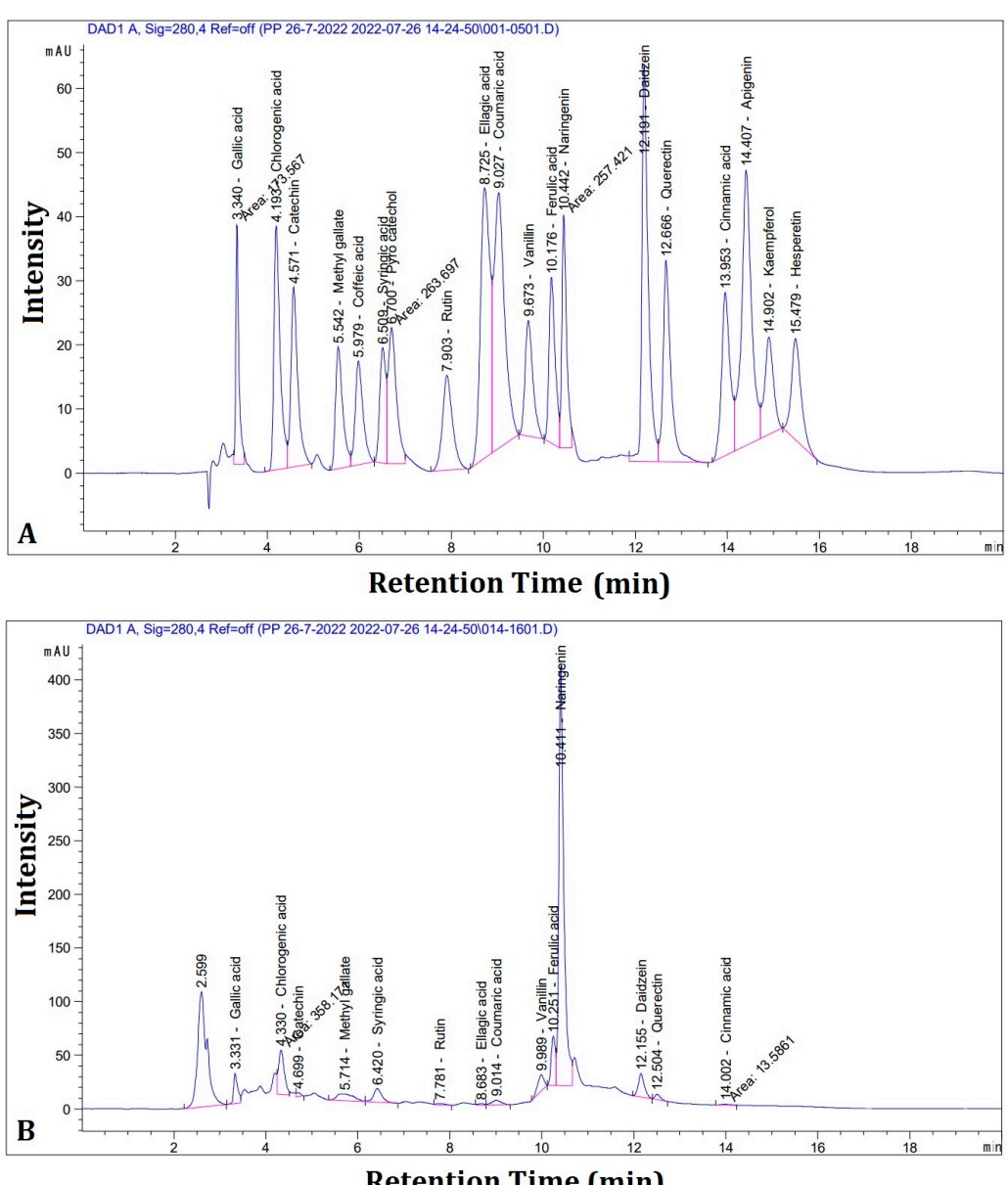

**Figure 4.** Phenolic and flavonoid standard compounds derived from HPLC instrument (**A**), and the polyphenolic content chromatograms detected in silk floss stem bark extract (**B**).

The authors of a study led by Soberón found that the *Candida albicans* fungus may be resistant to several antifungal agents. They extracted NAR from *Tessaria dodoneifolia* ethanolic extract, a plant used in Argentinean folk medicine, and evaluated its antifungal activity against two strains of *C. albicans* (ATCC 10231, susceptible to fluconazole, and 12–99, resistant to fluconazole). NAR displayed activity against the two strains (MIC, 40 µg/mL), and their results suggest that NAR combined with fluconazole can have a synergistic effect against the resistant strain [89]. However, at a concentration of 83 µg/mL, NAR had no antifungal activity against *Candida* spp. [90]. Differently, Vikram et al. [91] reported that NAR was ineffective at inhibiting the growth of 10231, *Aspergillus niger*, and *Saccharomyces cerevisiae* isolates [91]. Further investigation into the antifungal properties of NAR ether derivatives revealed that O-alkyl NAR derivatives and their oximes exhibited potent activity against *F. linii*, with naringenin, 7-O-dodecylnaringenin oxime, 7,4′-di-O-dodecylnaringenin, and their oximes all being capable of completely halting the growth of this particular fungal strain [86].

**Table 5.** Phenolic and flavonoid components detected in silk floss stem bark extract.

| Compounds | Area | Concentration (µg/g) |
|---|---|---|
| Gallic acid | 173.90 | 812.34 |
| Chlorogenic acid | 358.17 | 2727.49 |
| Catechin | 23.36 | 324.42 |
| Methyl gallate | 166.08 | 651.73 |
| Caffeic acid | * ND | ND |
| Syringic acid | 166.16 | 946.26 |
| Pyro catechol | ND | ND |
| Rutin | 15.57 | 97.16 |
| Ellagic acid | 13.11 | 139.18 |
| Coumaric acid | 62.34 | 104.47 |
| Vanillin | 145.02 | 473.22 |
| Ferulic acid | 285.48 | 1276.18 |
| Naringenin | 2968.31 | 18,698.83 |
| Daidzein | 195.35 | 628.91 |
| Quercetin | 37.72 | 220.42 |
| Cinnamic acid | 13.59 | 22.46 |
| Apigenin | ND | ND |
| Kaempferol | ND | ND |
| Hesperetin | ND | ND |

* ND = not detected.

Chlorogenic acid (CHA) is a type of phenolic compound found in various plants, like coffee and beans. It is highly regarded as a "gold plant" due to its ability to prevent microbial growth and act as an antioxidant, as reported by various studies [92–94]. Furthermore, CHA has been demonstrated to have inhibitory effects against *E. coli* and *B. subtilis* [92]. However, the specific mechanism by which it kills bacteria is not yet well understood. Several antimicrobial compounds with CHA as a component have the potential to enhance the permeability of the cell membrane and lead to the discharge of small molecules, as reported in studies conducted by Li et al. [95] and Zheng et al. [96]. Other studies indicate that CHA may have contributed to the release of intracellular proteins and ATP from *P. aeruginosa*, resulting in a compromised cell membrane as demonstrated by the penetration of bacteria cells, a stain for nucleic acids within cells treated with CHA, according to Su et al. [97] and Qian et al. [98].

Secondary metabolites, like gallic acid (GL) and ferulic acid (FC), have been studied for their antibacterial and antifungal properties [99,100]. A study examining their effects on *Listeria monocytogenes, S. aureus*, and *P. aeruginosa* revealed that FC was more efficient against the studied bacteria than GL and suggests that plant-derived chemicals can be a sustainable supply of innovative broad-spectrum antibacterial molecules. Hydrophobicity changes, negative surface charge decreases, and local rupture or pore growth in cell membranes generated by GL and FC leak essential intracellular contents. Some of the SFSB extract-detection molecules (e.g., coumaric acid, caffeic acid, ferulic acid, and α-tocopherol) were found to have antifungal activity against *Alternaria alternata* when applied alone or synergistically [100]. Our findings suggest polyphenolic compounds are crucial as antifungal molecules as the antimicrobial properties of the ethyl acetate extract of *C. insignis* may be due to flavones or their derivatives [101]. Various phenolic compounds in medicinal plants have been documented in prior studies and contribute to the plants' antimicrobial and antioxidant properties [102,103].

### 3.6. Silk Floss (SFSB) Extract GC-MS Analysis

The silk floss tree stem bark extract was analyzed using gas chromatography–mass spectrometry (GC-MS), and the results from Figure 5 and Table 6 revealed the presence of several compounds. Among them was cyclohexene, 1-methyl-5-(1-methylethenyl)-, (R)-, classified as a terpenoid with a relative abundance of 1.29%. Benzoic acid 2-(methylamino)-methyl ester, a benzoic acid derivative, was also identified, accounting for 1.32% of the

extract. Additionally, 1,2-benzenedicarboxylic acid, diethyl ester, a phthalate ester, was detected at a relative abundance of 0.73%. Terpenoids are known for their antimicrobial activity and have been reported to be effective against a range of microorganisms [104].

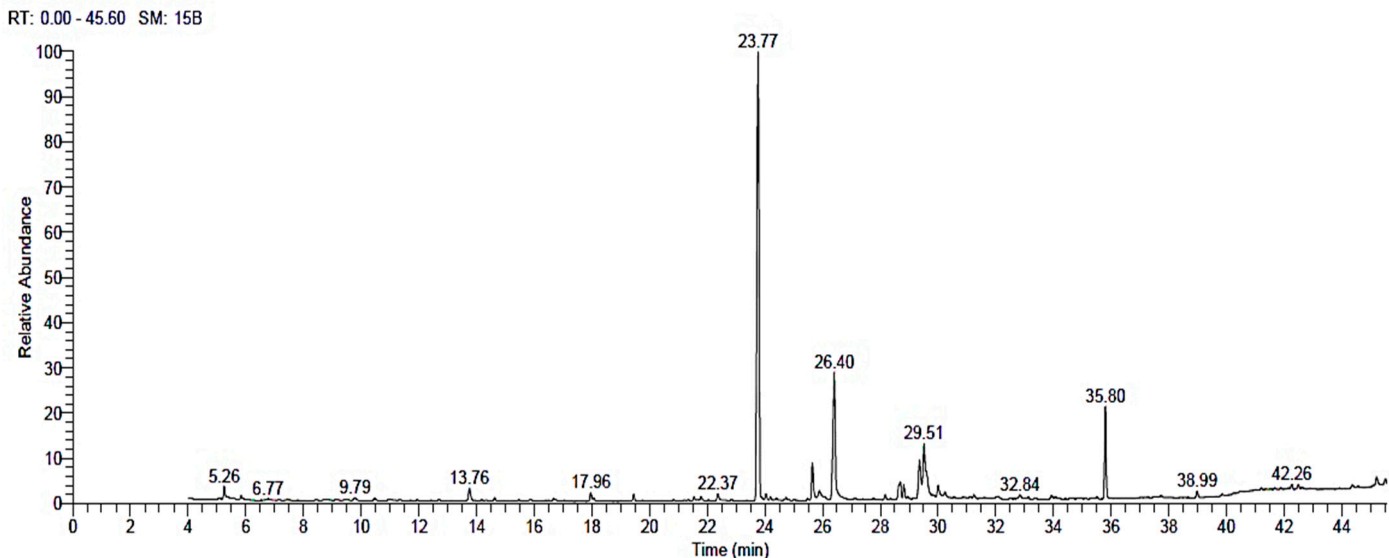

**Figure 5.** Phytochemical compounds identified in the GCMS analysis.

**Table 6.** List of GCMS compounds of the silk floss tree stem bark extract.

| RT | Compound | Class | Relative Abundance % |
|---|---|---|---|
| 5.25 | Cyclohexene, 1-methyl-5-(1-methylethenyl)-, (R)- | Terpenoid | 1.29 |
| 13.75 | Benzoic acid 2-(methylamino)-methyl ester | Benzoic acid derivative | 1.32 |
| 17.96 | 1,2-benzenedicarboxylic acid diethyl ester | Phthalate ester | 0.73 |
| 23.77 | Isopropyl myristate | Fatty acid ester | 55.61 |
| 25.64 | Palmitic acid methyl ester | Fatty acid ester | 3.51 |
| 26.41 | *n*-Hexadecanoic acid | Fatty acid | 14.05 |
| 28.64 | 12,15-Octadecadienoic acid methyl ester | Fatty acid ester | 1.09 |
| 28.70 | Linoleic acid methyl ester | Fatty acid ester | 1.42 |
| 28.82 | 11-Octadecenoic acid methyl ester | Fatty acid ester | 1.23 |
| 29.36 | 9,12-Octadecadienoic acid (Z,Z)- | Fatty acid | 3.95 |
| 29.52 | Oleic acid | Fatty acid | 4.65 |
| 29.60 | *cis*-Vaccenic acid | Fatty acid | 1.55 |
| 30.00 | Octadecanoic acid | Fatty acid | 1.17 |
| 35.80 | Bis(2-ethylhexyl) phthalate | Phthalate ester | 8.45 |

The dominant compound found in the extract was isopropyl myristate, a fatty acid ester, which constituted a significant relative abundance of 55.61%. This compound is widely utilized in cosmetics and personal care products due to its emollient properties. Another fatty acid ester, palmitic acid methyl ester, was present at a relative abundance of 3.51%. Fatty acids were also identified, including *n*-hexadecanoic acid, which accounts for 14.05% of the extract. Several fatty acid esters were detected, including 12,15-octadecadienoic acid methyl ester (1.09%), linoleic acid methyl ester (1.42%), and 11-octadecenoic acid methyl ester (1.23%). The extract also contained 9,12-octadecadienoic acid (Z,Z)-, a fatty acid with a

relative abundance of 3.95%. Furthermore, oleic acid (4.65%) and cis-vaccenic acid (1.55%), both fatty acids, were identified. Other fatty acids present in the extract were octadecanoic acid (1.17%) and palmitic acid (1.17%). Lastly, bis(2-ethylhexyl) phthalate, a phthalate ester commonly used as a plasticizer, was detected at a relative abundance of 8.45%. Fatty acids have been shown to possess antimicrobial properties, particularly against Gram-positive bacteria [105].

The data obtained from Table 5 provide valuable insights into the chemical composition of the silk floss tree stem bark extract. The presence of terpenoids, benzoic acid derivatives, phthalate esters, and various fatty acids suggests the potential presence of bioactive constituents in the extract. In a study conducted by Oke et al. [106], it was found that benzoic acid and its derivatives have demonstrated antimicrobial activity against various microorganisms. These compounds may have applications in various fields, including medicine, cosmetics, and industry. Further investigation and analysis of these compounds could help uncover their specific biological activities and potential uses.

## 4. Conclusions

In conclusion, this study highlights the partial antibacterial activity and complete suppression of fungal growth in *Rhizoctonia solani* by an ethanolic extract of silk floss stem bark (SFSB). The extract contains high levels of phenolic acids, flavonoids, terpenoids, benzoic acid derivatives, phthalate esters, and fatty acids with potential as a renewable solution for plant fungal infestations. Implementation of SFSB extract could offer an eco-friendly alternative to harmful bactericides, benefiting both human health and the environment. Future research should address the mechanism of antibacterial activity, field trials, non-target organism impact, and optimization of extraction processes for enhanced efficacy. Additionally, exploring synergistic effects, stability, and combined applications with other defense strategies will pave the way for sustainable agriculture practices.

**Supplementary Materials:** The following supporting information can be downloaded at: https://www.mdpi.com/article/10.3390/horticulturae9080912/s1, Figure S1: The inhibitory effect of silk floss tree stem bark extract (SFSB) at different concentrations (100, 200, 300, 400, 600, 800, 1000, 2000, and 3000 µg/mL) against various potato bacterial diseases; Figure S2: The inhibitory effect of silk floss tree stem bark extract (SFSB) at different concentrations (25, 50, 100, 200, and 300 µg/mL) on Rhizoctonia solani and Fusarium oxysporum fungal isolates.

**Funding:** The author would like to thank the Deputyship for Research and Innovation, "Ministry of Education" in Saudi Arabia, for funding this study (IFKSUOR3-402-1).

**Data Availability Statement:** Not applicable.

**Acknowledgments:** The author extends their appreciation to the Deputyship for Research and Innovation, "Ministry of Education" in Saudi Arabia for funding this research (IFKSUOR3-402-1).

**Conflicts of Interest:** The authors declare no conflict of interest.

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
