# Peer review of "Phytochemical Properties of Silk Floss Tree Stem Bark Extract and Its Potential as an Eco-Friendly Biocontrol Agent against Potato Phytopathogenic Microorganisms"

_horticulturae, doi:10.3390/horticulturae9080912_

Round 1
Reviewer 1 Report
1. Please reconsider the title, the results of this article were not strong enough to support the .. potential as an eco-friendly biocontrol agent....z
It just showed antimicrobial activity of the SFSB extract, but its activity was much lower when comparing to positive control and commercial agent comparison has not yet studied.
2. Table 3 and 4, please add standard deviation values and describe the meaning of statistical letters and there should be superscript.
3. Figure 1, I could not see the F. oxysporum FO-96 in phylogenetic tree. How was it different from FO-94?
4. Figure 2, y-axis should be % Inhibition or radical scavenging. Inhibition graph of standard BHT should be shown to compare with the extract activity.
5. Figure 3, please add chromatogram of the standard phenolic compounds.
Minor editing.
Reviewer 2 Report
Dear authors,
the search for eco-friendly biocontrol agents as alternatives to protective chemicals is very important and a global challange. In your manuscript you discribe the inhibiting effect of whole extratcs from the silk floss tree against several pathogenic species, both bacteria and fungi, which could be interesting for plant protection, especially in ecological farming. Thereby you focus a lot on diseases in potato with only few exceptions. In order to gain more visibility for this particular area of research, I suggest you add additional information to the title and to the keywords. I made associated comments in the pdf file in order to help you further improve your manuscript, please read them carefully.
The manuscript is well written and the results are indeed promising. In order to improve the current version for publication I made several comments, which you can find in the pdf file. First of all, you included a lot of relevant information and citations, which allows the reader to relate your research to other studies and connect them. However, I think that the way you present your results has to be improved before the manuscript can be published. For example, more information has to be provided in the caption of the figures, especially in the analytics part. The figures and captions need to include all relevant information that is necessary to understand the results presented. Please add additional information about the individual compounds, software, database, etc. Futhermore, the quality of figures needs to be improved as some figures look compressed and of low resolution.
Regarding the growth inhibition tests, it would be good to have some additional figures, which could also be provided as supplement, in order to visualize the results of the assays that have been preformed for bacteria and fungi (especially for R. solani, where you report a complete suppression of fungal growth).
Kind regards,
The reviewer

The manuscript is very well written but moderate editing of English language is still required, especially in terms of punctuation. I made some corrections in the pdf to help you with that but please check the entire manuscript again for errors.
Reviewer 3 Report
One of the major issues of the manuscript is the redaction of the methodology of each experimental part. Moreover, the presentation and analysis of the results.
Probably, the data of the manuscript could be published but it need more elaboration so that the reviewer and after the reader could understand the paper.
Reviewer 4 Report
the summary should be improved, following the typical guidelines one paragraph with introduction, another objectives, material and methods results and conclusion.
The introduction provides a well-organized and clear overview of the research, highlighting the significance of phytochemicals and plant extracts in combating plant diseases. However, to strengthen the introduction, specific objectives of the study should be explicitly stated.
The author claims that this study is the first to investigate the phytochemical composition and antimicrobial properties of the SFSB extract. While this is an exciting claim, it would be essential to support it with a comprehensive literature review to demonstrate that no other studies have explored this topic previously.
The "Materials and Methods" section is appropriate, with well-explained and organized procedures for identifying and isolating the phytopathogenic microorganisms. The section contributes to the study's reproducibility and reliability. The results are intriguing and present novelty in terms of inhibitory effects of the SFSB extract against various phytopathogenic microorganisms
The discussion can be improved
The conclusions need improvement, as they should go beyond summarizing the work. A well-structured conclusion should restate the main findings, discuss their implications, and suggest future research directions.
